# Primary Hepatocyte Isolation and Cultures: Technical Aspects, Challenges and Advancements

**DOI:** 10.3390/bioengineering10020131

**Published:** 2023-01-18

**Authors:** Impreet Kaur, Ashwini Vasudevan, Preety Rawal, Dinesh M. Tripathi, Seeram Ramakrishna, Savneet Kaur, Shiv K. Sarin

**Affiliations:** 1Department of Molecular and Cellular Medicine, Institute of Liver and Biliary Sciences, New Delhi 110070, India; 2School of Biotechnology, Gautam Buddha University, Greater Noida 201312, India; 3Department of Mechanical Engineering, National University of Singapore, Singapore 117581, Singapore

**Keywords:** primary hepatocytes, hepatocyte isolation, hepatocyte culture, organoids, liver on chip, perfusion bioreactor

## Abstract

Hepatocytes are differentiated cells that account for 80% of the hepatic volume and perform all major functions of the liver. In vivo, after an acute insult, adult hepatocytes retain their ability to proliferate and participate in liver regeneration. However, in vitro, prolonged culture and proliferation of viable and functional primary hepatocytes have remained the major and the most challenging goal of hepatocyte-based cell therapies and liver tissue engineering. The first functional cultures of rat primary hepatocytes between two layers of collagen gel, also termed as the “sandwich cultures”, were reported in 1989. Since this study, several technical developments including choice of hydrogels, type of microenvironment, growth factors and culture conditions, mono or co-cultures of hepatocytes along with other supporting cell types have evolved for both rat and human primary hepatocytes in recent years. All these improvements have led to a substantial improvement in the number, life-span and hepatic functions of these cells in vitro for several downstream applications. In the current review, we highlight the details, limitations and prospects of different technical strategies being used in primary hepatocyte cultures. We discuss the use of newer biomaterials as scaffolds for efficient culture of primary hepatocytes. We also describe the derivation of mature hepatocytes from other cellular sources such as induced pluripotent stem cells, bone marrow stem cells and 3D liver organoids. Finally, we also explain the use of perfusion-based bioreactor systems and bioengineering strategies to support the long-term function of hepatocytes in 3D conditions.

## 1. Introduction

The liver is primarily composed of two epithelial cell lineages, namely hepatocytes and cholangiocytes, which originate from hepatoblasts during fetal development. Hepatocytes are usually bi-nucleated and appear as sheets in the liver tissue. Isolated primary human hepatocytes are currently the gold standard for the development of human-relevant in vitro liver cell culture models. Despite their prolific growth ability in vivo, maintaining long-term cultures and proliferation of functional adult hepatocytes in vitro has been challenging. Isolated primary hepatocytes tend to differentiate into mesenchymal phenotypes after a few days in culture [1,2,3]. Apart from these Interindividual differences, variations in sample types, isolation procedures and other technical limitations have made hepatocyte isolation and culture a daunting task. Several researchers worldwide have long been optimizing hepatocyte isolation and culture protocols to allow them to grow steadily in vitro for downstream applications. Among these came the usage of immortalized cell lines like HepaRG, for instance, to achieve long-term culture of the hepatocytes, yet these cell lines cannot completely mimic the native functions of the hepatocytes in vivo [4,5]. With the onset of the 20th century, hepatocyte culture witnessed immense advancements, such as spheroid cultures [6], matrigel culture, tissue engineering-based scaffolds and 3D bio-printing to maintain long-term functional hepatocytes in vitro [7,8,9,10,11,12].

In this review, we discuss the conventional methods of hepatocyte isolation, along with the advancements achieved in the same.

## 2. Isolation of Primary Hepatocytes

Mechanical methods were employed previously to isolate hepatocytes (methods such as shaking with glass beads and filtering the liver through a cheesecloth), which caused extensive damage to cell membranes and the loss of function of the isolated hepatocytes. Later, due to the low yield of hepatocytes (5–10%) by the mechanical method, an enzymatic perfusion method was introduced when collagenase and hyaluronidase perfusion was carried through the liver via the portal vein in rats [13]. The enzymatic perfusion process greatly improved the yield and integrity of hepatocytes as the enzymes interacted with the majority of the cells through the liver vessels [14]. As an alternative to perfusion, slicing of rat liver tissues followed by collagenase digestion was also undertaken, giving intact and viable cells [15,16]. The enzymatic method was modified by Seglen, who pioneered a two-step perfusion procedure for liver cell isolation [17]. In this procedure, the rat liver is perfused with a Ca^2+^-free buffer, followed by perfusion with a collagenase buffer containing Ca^2+^. The isolation of primary rat hepatocytes in 1976 by a two-step procedure using collagenase was a major breakthrough [18,19,20,21]. The two-step method comprises of an initial perfusion with a calcium-free buffer to disrupt desmosomes that form the tight junctions between cells followed by a second perfusion with a calcium-rich buffer containing collagenase to further digest cell junctions. During the first step, desmosomes undergo irreversible structural changes with complete loss of cell–cell linkages which are unable to reform when the tissue is perfused with a second perfusion solution containing high concentrations of Ca^2+^, required for collagenase activity. The second step of the perfusion process uses collagenase to digest the ECM proteins and break cell–matrix contacts. 

Isolation of hepatocytes from resected liver tissues is the same as those of whole liver perfusion (Figure 1). Primary human hepatocytes are usually isolated from whole livers (not used for orthotopic transplantation) or resected liver tissues. A high flow-capacity perfusion device with a perfusion rate of up to 1–2 L/min is used for perfusion and isolation of hepatocytes from an adult human liver. Because there may be many openings of blood vessels on the cut surface of the liver, multiple perfusion cannulae and ligation of bypassing vessels are beneficial for maximal perfusion which increases yield and viability of the hepatocytes. Warm ischemia of liver tissues before starting the hepatocyte isolation procedure is an important factor that governs hepatocyte viability and yield. More than one hour of warm ischemic time is detrimental for achieving intact hepatocytes with high yields. Lee et al. used a two-step collagenase perfusion method in human livers to isolate primary hepatocytes. They started with a piece of liver with an intact Glisson’s capsule on all surfaces except for one cut surface. The study reported a large yield of hepatocytes with a cell viability of 77 ± 10% [22]. Some factors that have been suggested to influence the viability of isolated primary human hepatocytes include the presence of fibrosis, steatosis and bilirubin content. However, as perfusion may require large human liver tissue, with accessible vessels for cannulation, only a limited number of human tissue samples may be suitable for successful primary hepatocyte isolation. 

In this regard, a methodology to isolate human hepatocytes from resected healthy human liver tissue which is otherwise unsuitable for perfusion has also been reported. This method utilises similar buffers to the two-step perfusion method (EGTA and collagenase IV) and requires RBCs to be lysed following tissue digest to prevent contamination of RBCs in hepatocyte preparations. This method has been reported to obtain >65% viable hepatocytes despite using smaller liver tissue pieces. The study showed that the presence of a positive correlation between cell viability and liver tissue weight with >50 g [23]. This method of using only smaller liver fragments, instead of the whole tissue, allows cost-efficient scale-up of the procedure and also permits isolation of human hepatocytes without perfusion. 

Concerning enzymes for digestion, several collagenases have been tested for hepatocyte isolation. Class I collagenase enzymes are more stable and have a greater activity toward insoluble collagen, whereas class II enzymes are characterized by the ability to attack a significantly higher number of smaller peptides than their class I counterparts [24]. A combination of both classes of collagenases is important, but in particular, the presence of high amounts of class II collagenases represents an advantage for the isolation of hepatocytes [25]. Type IV is isolated from Clostridium histolyticum and has lower tryptic activity levels that limits the damage to membrane proteins and receptors but has normal to above normal collagenase activity when compared to other collagenase types and also results in a good yield of hepatocytes. Type IV is isolated from Clostridium histolyticum and is prepared to contain lower tryptic activity levels to limit damage to membrane proteins and receptors but has normal to above normal collagenase activity when compared to other collagenase types have also resulted in a good yield of hepatocytes.

Collagenase H and P prepared from the extracellular culture filtrate of Clostridium histolyticum are a mixture of different proteolytic enzymes and have high collagenase activity and have been mostly used for hepatocyte isolation protocols. Collagenase P (Roche, Mannheim, Germany) from Clostridium histolyticum (EC 3.424.3) was initially developed to isolate pancreatic islet cells but has also been described to be suitable for human liver tissues, especially when it is mildly fibrotic. Collagenase P containing perfusion solution has been supplemented with 10% fetal calf serum (FCS) that allows for prolonged but mild tissue digestion, resulting in a high yield of both parenchymal and non-parenchymal cells (NPCs) from a single liver tissue. Despite the type of collagenase used, in the end, the reconstituted enzyme solution should be sterile-filtered, and the dissolved enzyme should also be warmed up to 37 °C as collagenase rapidly loses its activity below this temperature.

Other modifications of hepatocyte isolation protocols include adaptations for parallel isolation and separation of NPCsof the liver. For example, in another approach, density gradient separation of NPCs in iodixanol and magnetic-activated cell sorting has been used for the efficient separation of individual NPC populations. Briefly, the protocol relies on the in situ perfusion of the liver through the portal vein. Initially, the peripheral blood is flushed from the liver with the help of a Ca^2+^-free buffer, which is followed by perfusion with collagenase digestion solution. Afterward, the liver is carefully removed and mechanically dissociated. Hepatocyte fraction is collected as a pellet at low-speed centrifugation and the supernatant is enriched in the NPCs. These NPCs containing supernatant are further subjected to density gradient centrifugation with Percoll at a high speed and individual NPCs (namely stellate cells, Kupffer cells and endothelial cells) are separated [26]. 

## 3. 2D Culture Models for Primary Hepatocytes

To mimic the micro-environment of hepatocytes in the liver in which they are surrounded by a low-density extracellular matrix (ECM) in the space of Disse, after isolation, hepatocytes are cultured on collagen-coated tissue culture wells or flasks. A healthy liver ECM is majorly comprised of type I and IV collagen, and fibronectin. Collagen-coated plates maintain the hepatocytes in culture for longer periods of time than the uncoated plates. Along with collagen, the effects of other ECM components have also been investigated on the growth of cultured hepatocytes [27,28]. Hepatocytes cultured on laminin-rich collagen gel, called matrigel, grow as round cells with significantly increased liver-specific functions for an enhanced time period than collagen [29]. Dunn et al. have cultured hepatocytes between two collagen hydrogel layers in a sandwich configuration. In this setting, hepatocytes maintain a high level of hepatic functions, such as albumin secretion and cytochrome P-450 activities. Sandwich cultures of human hepatocytes can stay for a prolonged period, that is, for >4 weeks [30]. Sandwich cultures can be successfully obtained from freshly isolated human or rat hepatocytes, as well as from cryopreserved counterparts. Transcriptional profiling showed that hepatocytes cultured between two layers of collagen had an enhanced expression of genes associated with energy metabolism and bile acid metabolic pathways. Sandwich cultures of human hepatocytes on collagen can also be performed in 96-well formats [31] which thus allow small to medium throughput approaches for various applications. Sandwich cultures of hepatocytes have also been established using a mix of collagen and matrigel in different combinations. Sandwich-cultured hepatocytes with an underlay of collagen seem to represent the best in vivo tissue architecture in terms of the formation of trabecular cell arrangement. Cultures overlaid with collagen were characterized by the formation of abundant bile canaliculi, while the bile canaliculi network in hepatocytes cultured on a layer of matrigel and overlaid with collagen showed the most branched and stable canalicular network [32]. The study suggested that the morphology and the multicellular arrangement were essentially influenced by the underlying matrix, while bile excretion and leakage of sandwich-cultured hepatocytes were mainly influenced by the overlay matrix. Heparin is another ECM element abundant in the liver. In a study by Foster et al., functional hepatocyte cultures were successfully maintained in a heparin-PEG gel sandwich culture system for over three weeks. The heparin gel layers were fabricated via UV light-induced thiol-ene coupling reaction of thiolated heparin (Hep-SH) and diacrylated poly (ethylene glycol) (PEG-DA) precursors. Analysis of hepatic function revealed that cells sustained albumin secretion for at least three weeks at similarly high levels for collagen and heparin gel sandwich culture systems. Furthermore, cytochrome P450 activity of hepatocytes was ~1.6 times higher in heparin gels compared to collagen gels. Another distinct advantage of heparin is its ability to bind and release growth factors in a controlled manner. It was observed that after 24 days, hepatocytes cultured in heparin gel sandwich containing hepatocyte growth factor (HGF) produced ~2 times higher amounts of albumin as compared to cells in collagen sandwiches [33].

## 4. Strategies to Improve Hepatocyte Cultures

### 4.1. 3D Cultures of Hepatocytes

In conventional 2D systems, long-term culture of primary hepatocytes undergo dedifferentiation and also trans-differentiation into mesenchymal lineages, thus leading to a loss of its polarity and functions. In this regard, several improvements in hepatocyte cultures have been experimented. Studies have focussed on creating the most native niche for these cells which has appropriate components of the growth factors, ECM and neighbouring cells allowing cell–cell contacts and cell–ECM interactions. Development of newer biomaterials to provide components of liver ECM, use of decellularized liver ECM, cultivation of hepatocytes as spheroids and organoids and use of perfusion culture systems have resulted in significant improvement in the viability and functionality of hepatocytes in ex vivo conditions. 

Three-dimensional aggregates of hepatocyte spheroids or organoids are composed of their self-organized cell clusters which do not attach to the substrates. These aggregates can be generated by using low-attachment plates, hanging drop cultures, use of microwells, matrigel cultures, rotating bioreactors, polymeric scaffolds, and so forth. For example, Chang et al. used a rotating wall vessel to culture primary mouse hepatocytes in a spheroid format and demonstrated that hepatic phenotype and gene expression were upregulated in the 3D format compared to monolayer 2D cultures [34]. Wong et al. used microwells to form primary hepatocyte spheroids, termed hepatospheres that hold therapeutic potential in regulating and repairing the damaged or diseased liver tissues. In hanging drop cultures and low attachment plates, hepatocyte spheroids exhibited long-term functions for up to five weeks [35]. Rose et al. developed the “hepoid” model of culturing primary human hepatocytes using both low attachment plates and collagen cultures. In this model, they seeded the cells into a low attachment plate and then included them in the collagen matrix. They showed that hepatocyte clumps remained grouped into clusters in the collagen matrix. Intriguingly, the conditions induced two waves of proliferation in the hepatocytes attributable to cell–cell and cell–matrix interactions. The cells-maintained polarity and exhibited hepatic differentiation and detoxification functions for 28 days [36]. 

Rebelo et al. introduced another unique culture technique for the differentiation of HepaRG under DMSO-free conditions, based on 3D culture and alginate encapsulation. This method combined HepaRG aggregation in spinner vessels with alginate encapsulation [37]. The study reported a highly polarized phenotype of hepatocytes with an interconnected network of bile canaliculi, inducible CYP3A4 and CYP1A activity, and functionality of phase III metabolism enzymes. MacPherson et al. used a nanofibrous hydrogel as a scaffold for the culture of human primary hepatocytes as 3D spheroids. The system was based on the cooperative assembly into a nanofibrous gel at the physiological pH of a fiber-forming peptide component, fluorenylmethyloxycarbonyl-diphenylalanine (Fmoc-FF), and an integrin-binding functional peptide ligand, Fmoc-arginine-glycine-aspartic acid (Fmoc-RGD) [38]. In our recent study, we used dipeptide (Dp) Isoleucine-α, β-dehydrophenylalanine (IΔF) blended with soluble liver extracellular matrix (sLEM) to form a hybrid scaffold (LEM1-Dp). Primary rat hepatocytes cultured on LEM_1_-Dp matrix grew in 3D format and maintained significantly higher cell viability and enhanced expression of albumin as compared to that observed in cells cultured on a collagen-coated substrate plate. Thus, peptide-based hydrogels provide a 3D biomimetic hepatic microenvironment [39]. 

Liu et al. established an enhanced human-simulated hepatic spheroid model system employing the human hepatoma cell line HepaRG grown in round-bottomed ultralow attachment 96-well plates with or without a decellularized liver biomatrix scaffold. The liver-specific matrix bioscaffold provided a liver-specific microenvironment and cells grown on these scaffolds showed better liver-specific functions (albumin secretion, urea synthesis, glycogen storage), cytochrome P450 enzymes (CYPs) metabolic activity, bile excretion, and increased expression of phase I and II metabolism enzymes, transporters, and nuclear receptors, and so forth as compared to those without these scaffolds [40].

Another approach to set up 3D cultures of hepatocytes is to initially mix them with matrigel to create hepatocyte organoids [41,42,43,44]. Similar to the spheroids, organoids also grow in 3D culture systems, but unlike spheroids, they are usually formed from resident hepatic stem cells/embryonic or induced pluripotent stem cells and partially resemble the whole organ, both in structure and function. With the advancement of organoid culture one can achieve indefinite expansion of self-renewing stem cells as well as adult cells into self-organizing 3D structures that resemble their native counterparts. Huch and colleagues described the culture of liver organoids derived from intrahepatic cholangiocytes, that is, the epithelial cells that form the bile ducts [45,46]. In a recent breakthrough, Nusse and Clevers group described the long-term organoid culture of murine and human primary hepatocytes [47,48]. Hu et al. established hepatocyte organoids using human hepatocytes from adult and paediatric sources. However, as compared to foetal-derived hepatocytes or adult mouse primary hepatocytes, the proliferative capacity of mature hepatocytes was limited (2–2.5 months). Gamboa and colleagues demonstrated that adult donor-derived hepatic stem cells and HepG2 cells in organoid culture conditions contained differentiated cells with hepatocyte-like functions. They also investigated different combinations of growth factors and small compounds to facilitate the expansion of adult donor-derived hepatic organoids and HepG2 cells. When the TGF inhibitor A8301 was present, the organoid expansion rates were significantly reduced, however they were noticeably increased when Forskolin (FSK) and Oncostatin M were present (OSM) [49].

Various methods of hepatocyte culture both 2D and 3D have been summarized in Figure 2, along with the advantages and disadvantages of each type.

### 4.2. Co-Culture of Hepatocytes with NPCs

Recent studies show that both cell–matrix and cell–cell interactions, as well as soluble factors, are vital for maintaining the functions of liver cells in vitro. Hence, culturing of hepatocytes along with other NPCs is always better for long-term maintenance of hepatocytes. Hepatocytes and other types of cells can be co-cultured in both random and micropatterned fashion patterns. Both the hepatic numbers and function have been reported to improve as a result of the co-culturing with the liver NPCs. Guguen-Guillouzo et al., in the 1980s, carried out some of the earlier studies that demonstrated improved hepatic function in the presence of NPCs [50]. Co-culturing hepatocytes has been performed either with sinusoidal cells or diploid human fibroblasts [51]. To achieve the optimal “co-culture effect”, the addition of corticosteroids to the culture media is essential, suggesting that certain soluble elements are also required for the successful operation of this model.

Bhandari et al. demonstrated that the presence of 3T3 fibroblasts led to higher markers of liver function, such as albumin and 7-ethoxyresorufin O-dealkylation (EROD) activity [52]. These results suggest that the presence of 3T3 fibroblasts led to enhanced vitality. According to the findings of Cho et al., however, the synthesis of albumin by primary hepatocytes that were co-cultured with 3T3-J2 fibroblasts was comparable to the that of hepatocytes that were cultured in collagen gel sandwiches [53]. Other groups have shown that the presence of primary hepatic stellate cells improves the function of hepatocytes by causing albumin secretion to be more sustained and higher [54,55,56]. In the late 1990s, Bhatia et al. set another development of hepatocyte co-cultures. They produced a pattern of collagen islands using photolithography for hepatocyte cultures and then cultured fibroblasts surrounding these hepatocytes [57]. Bell et al. incorporated functional NPCs into 3D spheroids containing primary human hepatocytes (PHH) and used this model (2:1 PHH: NPC) to examine acetaminophen (APAP) toxicity. The ability to create both inflammation and steatosis in NPC-containing spheroids demonstrates the potential utility of co-cultures for studying complex liver diseases [58]. 

Recently, extrusion-based 3D bioprinting has been used to precisely layer both hepatocytes, NPCs, and also ECM components in the desired configuration using different bioinks. Taymour and colleagues used 3D bioprinting to develop a core–shell strand/scaffold design and created a functional co-culture model of hepatocytes and fibroblasts. The 3D microenvironment of hepatocytes can be regulated and modified to simulate certain conditions by making specific adjustments to the composition of the bioink and by employing coaxial printing of two different bioinks in a core–shell fashion [59,60].

Major milestones in isolation and culture of the primary hepatocytes over the years have been summarized in Figure 3.

### 4.3. Liver-on-Chip Models for Hepatocyte Cultures

Because of their lack of physiological relevance, conventional in vitro cultures of hepatocytes do not impart optimal functions to the hepatocytes. The liver-on-a-chip technology has evolved to fully replicate the liver’s microarchitecture and in vivo physiological environment. In vivo and in vitro, oxygen availability is crucial for cell survival and function (Figure 4). This is particularly true for primary human cells, which rely heavily on oxidative phosphorylation (OXPHOS) for energy production. PHH has an oxygen consumption rate that is 12 times that of the regularly used hepatoma cell line HepG2/C3A and nearly 5 times that of HepaRG cells. Microfluidic and organ-on-a-chip systems aim to address this cellular requirement by injecting perfusion, a forced convection flow of medium to improve dissolved oxygen delivery to cells [60]. 

Allen et al. used a perfusion bioreactor system to co-culture primary rat hepatocytes and NPCs. The design supplied oxygen into the system via an inlet reservoir and measured the concentration of oxygen at the output by varying the flow conditions. It was seen that phosphoenolpyruvate carboxykinase activity was prevalent in the oxygen-rich zones upstream, and in the low-oxygen zones found downstream, cytochrome P450 family 2 subfamily B (CYP450 2B) activity was dominating. Cell death caused by acetaminophen hepatotoxicity was seen at the low-oxygen outflow [65]. In another study, Usta and colleagues discovered a zone-dependent glucose and nitrogen metabolism. These instances of re-creating the liver zonation by manipulating the oxygen gradient with a microfluidic approach are representative of how microfluidics can aid in the replication of the in vivo tissue environment [66]. Another factor that is most important for hepatocyte cultures is shear stress. Continuous blood flow exerts shear stress on hepatocytes and NPCs, which profoundly affect their functions [67]. Studies involving microfluidic devices with shear stress and hepatocytes is given in Table 1 and studies that have used perfusion-based dynamic liver on-chip microfluidic devices to fully replicate the liver sinusoids are given in Table 2. 

Shuler’s group used primary human hepatocytes and NPCs together in an environment with gravity-based flow. The system was made up of two layers of polydimethylsiloxane (PDMS). Each layer had a microchannel in it. A polycarbonate membrane was used to separate the microchannels. On the 3D scaffold, primary human hepatocytes and NPCs were grown together and joined to the membrane [75]. However, in this setup, spatial arrangement was not produced. Further, to overcome this problem, Prodanov et al. made two PDMS layers with microfluidic channels on them. A polyethylene terephthalate membrane was used to divide the microfluidic channels. The bottom channel was filled with primary hepatocytes and LX-2 cells, which are a human hepatic stellate cell line [76]. Jang et al. emulated human and cross-species drug toxicity using their microfluidic liver chip. The chip was made up of an upper parenchymal channel and a bottom vascular channel separated by a porous membrane. Primary hepatocytes from rats, dogs, and humans were cultivated in the top channel, while liver sinusoidal endothelial cells, Kupffer cells, and hepatic stellate cells were cultured on the lower channel’s porous membrane. Species-specific medication toxicities, as well as species-specific variances in the drug response, were simulated between humans and animals using a variety of phenotypes, including hepatotoxicity, steatosis modeling, and fibrosis [77]. Lee et al. used PDMS to create a concave microwell array. When compared to plane surface and cylindrical microwells, the concave microwell array aided in the creation of uniform-sized spheroids. The scientists observed that primary hepatocytes and hepatic stellate cell-based hepatospheres secreted 1.2-fold more albumin than primary hepatocyte-based hepatospheres [35]. Using a droplet microfluidic technique, Bhatia et al. encapsulated aggregated primary human hepatocytes and 3T3-J2 murine fibroblasts in PEG-DA. The encapsulated cells were subsequently captured using C-shaped traps and cultivated for 28 days under perfusion conditions [78].

Lin et al. created an artificial hepatic blood flow as well as an artificial bile flow in their experiment. The chip was composed of three layers, which were separated by two membranes made of polycarbonate. When exposed to hepatotoxicity-inducing compounds, cells grown in this technique demonstrated greater sensitivity to hepatotoxicity than cells grown in well plate-based static cultures [79]. The Groger group created a microchip MOTiF biochip vascular organic-organoid model. The sinusoid chip was used to culture human umbilical vein endothelial cells (HUVEC), monocytes, HepaRG, and stellate cells [80]. Long’s group created an in vitro liver sinusoid chip by combining four different types of primary murine hepatocytes into two adjacent fluid channels separated by a porous permeable membrane, thus recreating the liver’s main structures and topologies. The chip was treated with LPS (lipopolysaccharide) to cause inflammation, according to the authors. The presence of neutrophils in the chip was confirmed as an inflammatory response to LPS [81]. Lee et al. created a 3D model to study alcohol liver disease using a spheroid-based microfluidic system with both hepatocytes and NPCs. The authors assessed changes in hepatocyte function on the chip after ethanol exposure. The roughness of the hepatocyte spheroid surface grew as the concentration of ethanol exposed to the cells rose, and cell viability dropped. Rainer et al. used free fatty acids (FFA), palmitic acid and oleic acid, to induce steatosis in the chip where the liver sinusoid was architecturally implemented. The chip culture accumulated triglycerides at a slower pace than the 2D-well plate culture. This may be a more accurate representation of the in vivo condition of chronic steatosis than standard in vitro models [82]. Another model replicated non-alcoholic fatty liver disease (NAFLD) using LiverChip^®^, with 3D cultures of hepatocytes in a collagen scaffold. They were subjected to FFA in order to produce steatosis, and the reduction of fat was proven by employing therapeutic drugs such as pioglitazone and metformin. The authors of this study were able to demonstrate that a disease model based on a chip can be used to investigate the impact that a variety of medications on hepatocyte functions [83]. An in vitro model of the progression from the early inflammation (NASH) to the fibrosis of NAFLD was developed by Sung et al. in their study. This model was established on a chip that was capable of cultivating hepatocytes and endothelial cells in a 3D format shape using gelatin hydrogel. The researchers found that hepatocytes and endothelial cells cultivated in a 3D shape successfully developed into their respective cell types. For inducing NAFLD, palmitic acid was added to the medium, and for generating NASH, TGF-beta was added to the medium that was used to produce NASH [84].

## 5. Media Components in Hepatocyte Cultures

Irrespective of the culture conditions, the culture media used for hepatocytes are generally Dulbecco’s modified Eagle medium (DMEM), Williams’ medium E or modified Chee’s medium, supplemented notably with insulin and glucocorticoid like dexamethasone or hydrocortisone. The use of fetal bovine serum [10% (volume/volume)] is not advised for hepatocytes. It can, however, be used for the initial period of hepatocyte seeding (24 h) to promote attachment. FBS has been reported to disrupt the formation of bile canaliculi, and hence, bile secretion [85], and is also known to promote hepatocyte de-differentiation. Hence, a serum-free medium is preferred along with supplements, such as Endothelial Growth Factor (EGF), insulin, glucocorticoid hormone and glucagon that help to maintain the hepatocytes in their differentiated state. The ability of EGF to induce DNA synthesis in primary hepatocytes was first demonstrated in 1976. Additionally, studies have shown that HGF also promotes DNA synthesis [86]. HGF is in fact known to be a key factor in enhancing boosting urea formation. Another factor, oncostatin M (OSM) induces hepatic maturation and differentiation in a primary culture of embryonic stem cells or induced pluripotent stem cells [87]. Besides this, insulin has been reported to improve cell morphology [88] and glucocorticoids, such as dexamethasone or hydrocortisone, induce hepatic gene expression [89]. Another component essential for initiating hepatocyte proliferation is Wnt3a, as removal of Wnt3a has been reported to substantially reduce the proliferation of human hepatocytes [90].

## 6. Characterization of Primary Hepatocytes

A thorough characterization of the isolated hepatocytes is essential to ensure that functions of interest are maintained during cell cultures. Several tests exist for assessing their phenotype and functionality. The morphology of hepatocytes in a 2D culture is characterized by a polygonal shape and bi-nucleated appearance that is comparable to epithelial cell cultures [91]. The investigation of morphology in combination with albumin secretion is frequently used as proof of hepatocyte functionality. Albumin and Urea secretions from cultured cells are the most widely used characterization markers for the primary hepatocytes since the 1900s [92]. With advancements in culture techniques, further validation of the isolated hepatocyte’s functions is being done by xenobiotic metabolizing phase I and phase II CYP enzyme activities, glucose metabolism and ammonia detoxification [93,94]. Various inducers such as rifampicin, phenobarbital, beta-naphthoflavone are widely used to measure the CYP enzyme activity of the hepatocytes post-treatment, in terms of (a) gene expression through RT-PCR [95]; (b) luciferin-based kits that can specifically bind to particular CYP enzymes [96,97,98]; (c) NGS-based arrays to determine ADME properties [99]. 

Enzyme activities such as Aspartate transaminase (AST) and Lactate dehydrogenase (LDH) are also being estimated to characterize hepatocytes in 2D and 3D culture conditions. Proteomic and transcriptomic analysis has also contributed to the identification of novel characterization markers of hepatocytes in 3D culture environments [100,101,102,103].

Figure 5 shows the schematic representation of the methods involved in characterization of isolated primary hepatocytes. 

## 7. Liver Zonation: Challenges and Future Directions 

The field of liver bioengineering has grown substantially in the past few years, proving novel biomaterials, multiple cell types and microfluidic devices for improved cultures of primary hepatocytes. However, in order to achieve optimal hepatocyte function in vitro, multiple liver cell types have to be incorporated in the correct spatial orientation such that the cells can have a maximal effect on hepatocyte functions. Liver can be divided into three major zones depending on its position from the portal triad. The hepatocytes situated adjacent to the portal vein and hepatic artery are termed as the periportal (zone 1), those that surround the central vein are known as pericentral hepatocytes (zone 3), and the cells in between these areas are called mid-lobular hepatocytes (zone 2) [104]. Similar to hepatocytes, liver sinusoidal endothelial cells and hepatic stellate cells can also belong to these three distinct zones [105,106]. Cells from different zones vary considerably in terms of gene expression and metabolic functions. The molecular signatures of liver cells in different zones is mainly affected by a gradient of oxygen and nutrients. Creating different zones of the liver ex vivo is a big challenge as there is mostly a uniform distribution of oxygen and other nutrients in culture systems. To date, there are very limited studies that have been able to re-capitulate different zones of the liver. Bioprinting with multiple print heads have let the researchers to achieve the exact spatial orientation of the hepatocytes along with the other NPCs found in the liver [107]. A study encapsulated different cell types such as, hepatocytes, human umbilical vein endothelial cells (HUVECs) and human lung fibroblasts (HLFs) in a collagen solution as the bioink in polycaprolactone (PCL) as the framework to fabricate a 3D microenvironment for hepatocytes. Co-culture with NPCs enhanced hepatocyte survival, protein secretion and metabolism of hepatocytes. However, this study did not create any distinct zonation patterns of hepatocytes. Recently, Janani et al. used a different approach for creating zonal hepatocytes. They showed that by using varying liver ECM-functionalized silk scaffolds along with bioreactor systems with an oxygen gradient, one can achieve zonal-based functions of the cultured hepatocytes. The study demonstrated that scaffolds with high ECM supports periportal-specific albumin synthesis, urea secretion, and bile duct formation, though scaffolds with low ECM supports pericentral-specific cytochrome P450 activity [108]. Another study showed how the effect of flow in a microchannel bioreactor system affects metabolic functions of primary rat hepatocytes [109]. Hepatocytes cultured with a low wall shear stress in the bioreactor (0.01 to 0.33 dyn/cm^2^) led to 2.6 to 1.9 times higher albumin and urea synthesis rates, as compared to hepatocytes cultured at higher wall stresses (5 to 21 dyn/cm^2^). Bioreactor systems with a gradient of flow and oxygen along with varying ECM composition might help to create different microenvironments similar to the native liver, thus allowing to capture the metabolic heterogeneity of hepatocytes in culture. Along with hepatocytes, we also need to seed Liver sinusoidal endothelial cells (LSECs) and Hepatic Stellate Cells (HSCs) in 3D co-culture models and then culture them in fluidic gradient culture systems to have all three major cell types arranged in an adequate spatial orientation giving rise to zonation of liver cells as well.

## 8. Conclusions

Hepatocytes are indeed one of the toughest cells to isolate and preserve under ex vivo conditions. They also cannot be maintained in hypothermic and cryopreserved condition for a long time. Maintaining these cells as 3D spheroids and organoids in liver-on-chip dynamic culture systems represents the most apt and ideal strategy for their prolonged cultures. Given the fact that cultured hepatocytes are highly relevant to drug screening, the study of viral hepatic diseases and regenerative therapies, we need ready-to-use and miniaturized sinusoid-like perfusion devices to not only maintain them efficiently in culture for extended periods, but also transfer them to the bedside of the patients for transplantation purposes.

## Figures and Tables

**Figure 1 bioengineering-10-00131-f001:**
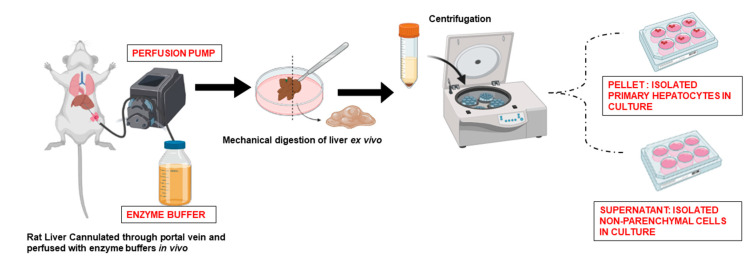
Schematic representation of the conventional method of isolation of primary hepatocytes and the non-parenchymal cell population of the liver through perfusion method. Original images by authors created with Biorender.com (accessed on 20 October 2022).

**Figure 2 bioengineering-10-00131-f002:**
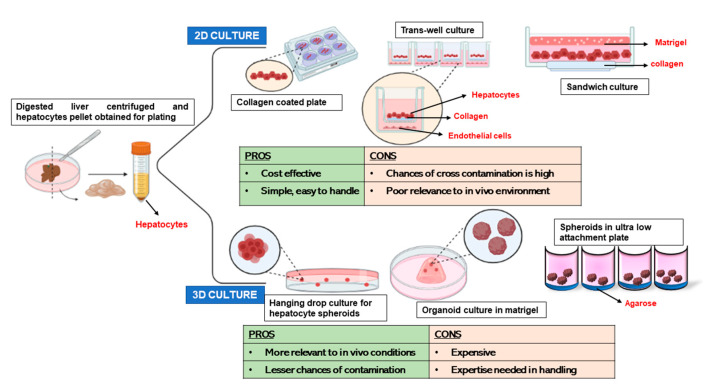
Various methods of 2D and 3D culture of hepatocytes. 2D methods include conventional collagen-coated plates, trans-well and sandwich culture and 3D culture methods include hanging drop culture, organoid culture and spheroid culture. Advantages and disadvantages of each method provided. Original images by authors created with Biorender.com (accessed on 20 October 2022).

**Figure 3 bioengineering-10-00131-f003:**
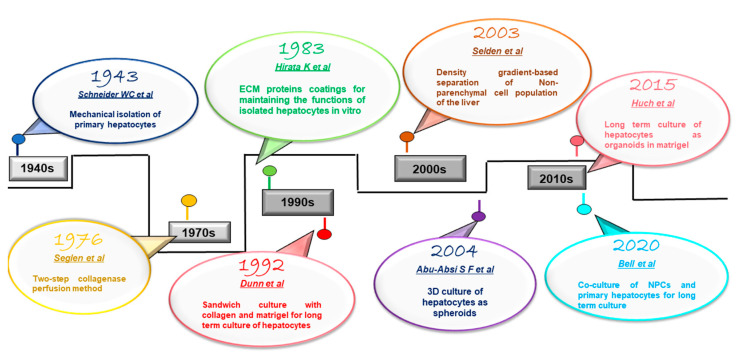
Major milestones in the isolation and culture of primary hepatocytes and non-parenchymal cell population of the liver [29,30,46,58,61,62,63,64].

**Figure 4 bioengineering-10-00131-f004:**
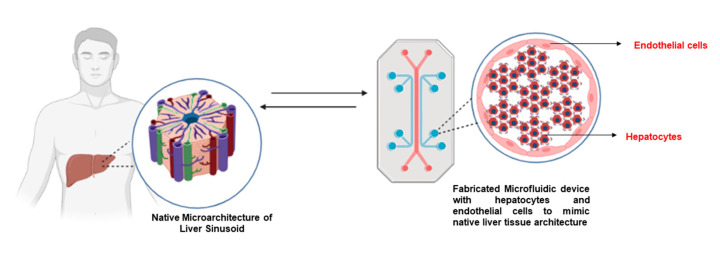
Schematic representation of a liver-on-chip device to mimic the native liver tissue microarchitecture. Original images by authors created with Biorender.com (accessed on 20 October 2022).

**Figure 5 bioengineering-10-00131-f005:**
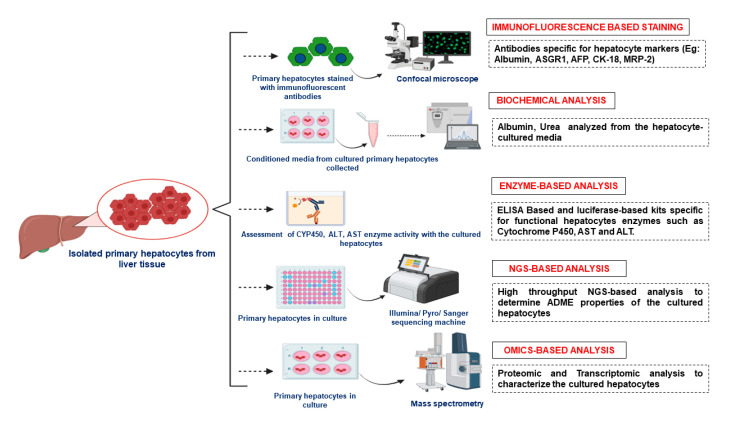
Characterization of the isolated primary hepatocytes using different techniques. ASGR1—Asailoglycoprotein Receptor1, AFP—Alpha Fetoprotein, CK18—Cytokeratin-18, MRP-2—Multi Drug Resistance Protein-2, ELISA—Enzyme Linked Immunosorbent Assay, AST—Aspartate aminotransferase, ALT—Alanine aminotransferase, ADME—Absorption, Distribution, Metabolism and Excretion.

**Table 1 bioengineering-10-00131-t001:** Studies that have used liver-on-chip configurations with shear stress to mimic the native liver sinusoid enlisted. (PSC-Pluripotent Stem Cells).

S. No.	Group	Shear Stress	Cell Type Used	Ref
1.	Tanaka et al.	1.4 to 60 dyne/cm^2^	HepG2	[68]
2.	Vinci et al.	5 × 10^−6^ dyne/cm^2^	Primary hepatocytes	[69]
3.	Rashidi et al.	2.9–4.7 × 10^−6^ dyne/cm^2^	Hepatocyte-like cells derived from human embryonic stem cells and induced PSCs.	[70]

**Table 2 bioengineering-10-00131-t002:** List of studies that have used liver-on-chip configurations with shear stress for hepatocyte cultures. (ECM-Extracellular matrix; PDMS-Polydimethylsiloxane).

S.No.	Group	Microfluidic Chip Specifications	Cell Type Used	Ref
1.	Lee et al.	2 µm in width, 1 µm in height, and 30 µm in length	Primary hepatocytes	[71]
2.	Toh et al.	3D ECM was formed by injection of methylated collagen and terpolymer hydroxylethylmethacrylate–methylmethacrylate–methylacrylic acid.	3D Culture of HepG2 and primary hepatocytes	[72]
3.	Goral et al.	Patterened microstructure of PDMS added to the bottom of the cell culture chamber serves as independent perfusion microchannels.	Primary human hepatocytes	[73]
4.	Banaeiyan et al.	Liver-lobule-like hexagonal tissue culture chambers having flow channels with flow rate 1 µL/min	HepG2 cells and human-induced pluripotent stem cell (hiPSC)-derived hepatocytes	[74]

## Data Availability

Not applicable.

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
