# Peer review of "Primary Hepatocyte Isolation and Cultures: Technical Aspects, Challenges and Advancements"

_bioengineering, 2023, doi:10.3390/bioengineering10020131_

Round 1

Reviewer 1 Report

Hepatocytes are very useful for investigating hepatic disease, but the cells were difficult to culture while maintaining their function. In the manuscript, the authors systemically describe the isolation of Primary hepatocytes, hepatocyte culture (2D, 3D, and liver-on-chip models), and characterization of primary hepatocytes.  Schematic representations were provided in most of the steps. It is very helpful and easy for to be leaned by audient. 

1. I think the “6. Liver-on-Chip Models for Hepatocyte Cultures” section belongs to the “4. Strategies to Improve Hepatocyte Cultures” topic. Please combine them together.

2. Characterization of Primary Hepatocytes is a very important and necessary step to validate the culture method. It is better to provide one schematic representation for this section

3. The reference format is not consistent, please revise it.

Author Response

Author response:

 We thank the reviewer for appreciating the manuscript and the figures. We have revised the manuscript and highlighted all the revisions made.

  1. As per the reviewer’s suggestion, “6. Liver-on-Chip Models for Hepatocyte Cultures” has been combined with 4. ‘’Strategies to Improve Hepatocyte Cultures’’ as section 4.3 (line no. 309-409).
  2. We have now included a figure for the characterization of primary hepatocytes as suggested by the reviewer (Figure 5: Characterization of the isolated primary hepatocytes using different techniques))
  3. We have revised the references as per the suggestions.

Reviewer 2 Report

The review describe the development and recent update on the hepatocytes isolation and culture with enough depth and width. However, the major difficulties/challenges in the establishment of 3-D hepatocyte culture with multiple cell types in a correct spatial distributions have not been fully discussed. Further the specific challenges, the roles and the research directions that the fields of bioengineering could focus to advance the field have not been discussed in the review.

Author Response

We thank the reviewer for this valuable suggestion. We have added a complete section on Liver Zonation: Challenges and Future Direction in the revised manuscript as follows:

The field of liver bioengineering has grown substantially in past few years proving novel biomaterials, multiple cell types and microfluidic devices for improved cultures of primary hepatocytes. However, in order to achieve optimal hepatocyte function in vitro, multiple liver cell types have to be incorporated in the correct spatial orientation such that the cells can have maximal effect on hepatocyte functions. Liver can be divided into three major zones depending on its position from the portal triad. The hepatocytes situated adjacent to the portal vein and hepatic artery are termed as the periportal (zone 1), those that surround the central vein are known as pericentral hepatocytes (zone 3), and the cells in between these areas are called as mid-lobular hepatocytes (zone 2) [100]. Similar to hepatocytes, liver sinusoidal endothelial cells and hepatic stellate cells can also belong to these three distinct zones [101,102]. Cells from different zones vary considerably in terms of gene expression and metabolic functions. The molecular signatures of liver cells in different zones is mainly affected by a gradient of oxygen and nutrients. Creating different zones of the liver ex vivo is a big challenge as there is mostly a uniform distribution of oxygen and other nutrients in culture systems. Till date, there are very limited studies that have been able to re-capitulate different zones of the liver. Bioprinting with multiple print heads have let the researchers to achieve the exact spatial orientation of the hepatocytes along with the other NPCs found in the liver [103]. A study encapsulated different cell types such as, hepatocytes, human umbilical vein endothelial cells (HUVECs) and human lung fibroblasts (HLFs) in a collagen solution as the bioink in polycaprolactone (PCL) as the framework to fabricate a 3D microenvironment for hepatocytes. Co culture with NPCs enhanced hepatocyte survival, protein secretion and metabolism of hepatocytes. However this study did not create any distinct zonation patterns of hepatocytes. Recently, Janani et al used a different approach for creating zonal hepatocytes. They showed that by using varying liver ECM-functionalized silk scaffolds along with bioreactor systems with an oxygen gradient, one can achieve zonal based functions of the cultured hepatocytes. The study demonstrated that scaffolds with high ECM supports periportal-specific albumin synthesis, urea secretion, and bile duct formation, albeit scaffold with low ECM supports pericentral-specific cytochrome P450 activity (104). Another study showed how the effect of flow in a microchannel bioreactor system affects metabolic functions of primary rat hepatocytes [105]. Hepatocytes cultured with a low wall shear stress in the bioreactor (0.01 to 0.33 dyn/cm2) led to 2.6 to 1.9 times higher albumin and urea synthesis rates, as compared to hepatocytes cultured at higher wall stresses (5 to 21 dyn/cm2). Bioreactor systems with a gradient of flow and oxygen along with varying ECM composition might help to create different microenvironments similar to the native liver, thus allowing to capture metabolic heterogeneity of hepatocytes in culture. Along with hepatocytes, we also need to seed LSECs and HSCs in 3D co-culture models and then culture them in fluidic gradient culture systems to have all three major cell types arranged in an adequate spatial orientation giving rise to zonation of liver cells as well.

Reviewer 3 Report

This is a nicely written review summarizing the current techniques to isolate and culture primary hepatocytes. The information contained in this review is comprehensive and well organized. The content is up-to-date and should be of interest to a broad readership.

Author Response

We thank the reviewer for his encouraging comments.

We have now thoroughly verified the manuscript for spelling errors as per the reviewer’s suggestion.